Research on path planning of mobile robots based on improved A* algorithm

Fu Xing 1 2
Huang Zucheng 1 zc.huang@giat.ac.cn
Zhang Gongxue 2
Wang Weijun 1
Wang Jian 1
1 Robot and Intelligent Equipment Center, Guangzhou Institute of Advanced Technology , Guangzhou, Guangdong , China
2 College of Mechanical and Electrical Engineering, Shaanxi University of Science and Technology , Xi’an, Shaanxi , China
Szénási Sándor
Electronic publication date: 2025 Feb 25
Publication date: 2025
Volume: 11
Electronic Location ID: e2691
Received 2024 Aug 23; Accepted 2025 Jan 19
Copyright: © 2025 Fu et al.
Copyright year: 2025
Copyright holder: Fu et al.
License: This is an open access article distributed under the terms of the Creative Commons Attribution License, which permits unrestricted use, distribution, reproduction and adaptation in any medium and for any purpose provided that it is properly attributed. For attribution, the original author(s), title, publication source (PeerJ Computer Science) and either DOI or URL of the article must be cited.
License URL: https://creativecommons.org/licenses/by/4.0/

Keywords: A* algorithm, Hybrid search, Path planning, Node deletion, Bidirectional search

Funding: National Key Research and Development Program of China 2018YFA0902900 This work was supported by the National Key Research and Development Program of China (No. 2018YFA0902900). The funders had no role in study design, data collection and analysis, decision to publish, or preparation of the manuscript.

==============================
To address the issues of low search efficiency, excessive node expansion, and the presence of redundant nodes in the traditional A* algorithm, this article proposes an improved A* algorithm for mobile robot path planning. Firstly, a multi-neighborhood hybrid search method is introduced, optimizing the traditional eight-neighborhood and twenty-four-neighborhood into a new sixteen-neighborhood. The choice between eight-neighborhood search and sixteen-neighborhood search is determined based on the presence of obstacles in the eight-neighborhood around the current node, effectively enhancing the search efficiency of the algorithm and reducing the number of nodes expanded during the search process. Subsequently, unnecessary nodes are eliminated based on the positional relationship between the current node and the target node, according to neighborhood direction search rules, further decreasing the number of expanded nodes. Additionally, improvements to the bidirectional search mechanism along with the incorporation of dynamic weight coefficients further enhance the search efficiency of the algorithm. Furthermore, a strategy for extracting key nodes is employed to effectively remove useless turn points, thus resolving the issue of redundant nodes. Finally, simulation experiments demonstrate that the proposed improved A* algorithm outperforms the traditional A* algorithm in terms of search speed, number of expanded nodes, and path length, validating the effectiveness of the proposed method.

Introduction

In recent years, with the rapid development of the automation and intelligent era, mobile robots have seen a significant surge in market demand and vast application prospects. They have been widely applied in numerous fields, including healthcare, industry, services, education, and entertainment (Hanxi et al., 2021; Rubio, Valero & Llopis-Albert, 2019; Rybczak, Popowniak & Lazarowska, 2024). Path planning (Sánchez Ibáñez, Pérez del Pulgar & García Cerezo, 2021; Promkaew et al., 2024) has consistently been one of the key technologies in the research of mobile robots. The primary goal of path planning is to swiftly and accurately find an optimal collision-free path from a starting position to a target position in an specific environment (Aggarwal & Kumar, 2020). Currently, methods for mobile robot path planning are primarily categorized into two main classes: traditional algorithms and intelligent algorithms. Traditional algorithms include global path planning algorithms and local path planning algorithms. Common global path planning algorithms include Dijkstra’s algorithm (Tang, Sheng & Sun, 2023; Alshammrei, Boubaker & Kolsi, 2022), A* algorithm (Hart, Nilsson & Raphael, 1968; Singh et al., 2018), D* algorithm (Ren et al., 2022), and rapidly-exploring random tree (RRT) algorithm (Haitao et al., 2023). Local path planning algorithms involve methods such as the dynamic window approach (Kobayashi & Motoi, 2022) and artificial potential field method (Bin & Ning, 2024; Weiguo et al., 2023). Intelligent algorithms encompass Ant Colony Optimization (Qingbin, 2024), particle swarm optimization (Phung & Ha, 2021), and genetic algorithms (Kobayashi & Motoi, 2022).While intelligent algorithms can be used for global path planning, their implementation in real-time scenarios is less common than algorithms such as A* or DWA (Israr et al., 2022).

Among global path planning algorithms, the A* algorithm is widely used in path planning research due to its simplicity in computation and relatively short planned paths. However, it also suffers from issues such as low search efficiency, long run times, excessive reliance on heuristic functions, and redundant waypoints. Many scholars, both domestically and internationally, have conducted improvement studies to address these problems. For instance, Yawen et al. (2023) proposed an improved A* algorithm capable of searching within a twenty-four neighborhood, which expands the traditional eight neighborhood search strategy and reduces the number of waypoints, achieving shorter paths; however, this resulted in an excessive decision-making time. Fangfang et al. (2024) introduced a modified A* algorithm based on adaptive environmental mapping that takes into account the global distribution of obstacles and incorporates adaptive obstacle strategies, effectively improving decision-making time, but its application is limited to known static environments. Changgeng (2022) proposed a bidirectional alternating search strategy for an improved A* algorithm, optimizing the cost function and effectively enhancing search efficiency; however, the rapid search led to paths that were substantially longer than those generated by the traditional A* algorithm. Rongshen et al. (2024) combined an improved A* algorithm with the Dynamic Window Approach for path planning, reducing the traditional eight neighborhoods to five and applying dynamic window techniques for path smoothing, effectively reducing redundant node counts and path length, though the complexity of the algorithm is high and efficiency is relatively low. Min et al. (2021) incorporated the influence cost of curvature paths into the cost function of the traditional A* algorithm, reducing the number of turns in the path and significantly improving smoothness; nonetheless, redundant nodes still persisted in the planned paths. Tang et al. (2018) proposed an improved hybrid A* algorithm, integrating the concepts of artificial potential fields, effectively reducing maneuvers and the maximum path curvature, but with relatively low search efficiency. Duchoň et al. (2014) contributed an improved A* algorithm based on RSR and JPS, introducing Basic and Theta* parameters to effectively plan optimal paths, though its application scenarios are limited.

Based on the aforementioned research, the existing improvements to the A* algorithm for mobile robot path planning primarily focus on four areas: neighborhood search enhancements, heuristic function optimization, path curvature smoothing, and the integration of other algorithms. These studies have improved the performance of the traditional A* algorithm in addressing mobile robot path planning to varying degrees; however, their effectiveness in reducing unnecessary expanded nodes and enhancing algorithm efficiency has been limited. Therefore, this article proposes an improved A* algorithm specifically for mobile robot path planning. Firstly, the traditional eight neighborhoods and twenty-four neighborhoods are optimized and expanded into a new sixteen neighborhood. By determining whether obstacles exist in the eight neighborhoods surrounding the current node, the algorithm selects either the eight neighborhood search or the new sixteen neighborhood search approach, effectively reducing the number of node expansions and improving efficiency. Subsequently, unnecessary nodes are removed according to neighborhood direction search rules, further decreasing the expansion count during the search process. Furthermore, improvements are made to the bidirectional search mechanism to enhance the search efficiency of the algorithm. Additionally, a strategy to extract key nodes is implemented to effectively eliminate unnecessary waypoints, significantly addressing the issue of redundant nodes. Finally, the effectiveness of the proposed algorithm is verified through simulation experiments.

Problem description

Grid-based environment modeling

Environment modeling is a critical aspect in the study of path planning for mobile robots. An appropriate environmental map model can significantly enhance the search efficiency of algorithms. Commonly used modeling techniques include grid-based methods (Weikuan & Baoping, 2019), topological methods (Chong et al., 2006), and visibility graph methods (Xiaodan, 2021). Given the advantages of visualization and interpretability offered by the grid-based method, along with its ease of data manipulation during algorithm simulation and validation, this article adopts the grid-based method for environmental modeling.

To facilitate the effective positioning of the mobile robot, the path planning nodes are constrained to the centers of the grid cells. The robot is considered as a mobile point mass, with its position determined by the grid cells on the map. This approach effectively reduces the complexity of the path. Figure 1 illustrates a grid example of size 10 × 10, wherein the red grid cell represents the starting position of the mobile robot, the blue grid cell denotes the target position, the black grid cells indicate obstacles, and the white grid cells signify passable areas.

Figure 1 Grid map.

Once the grid model is established, each grid cell must undergo an encoding process. Each grid cell can have one of two states: vacant or occupied. When the encoding of a grid cell is set to 1 (i.e., when the color of the cell is black), it signifies that the grid is in an occupied state, indicating the presence of an obstacle. Conversely, when the encoding is set to 0 (i.e., when the cell is white), it denotes that the grid is vacant, allowing the robot to pass through without obstruction.

Traditional A* algorithm

The traditional A* algorithm is a heuristic approach that combines the advantages of both Dijkstra’s algorithm and the Breadth-First Search (BFS) algorithm, effectively addressing the pathfinding problem. Once the initial point is determined, the algorithm employs a heuristic search method. Specifically, it calculates the cost associated with each reachable node, which is the sum of the cost incurred to reach that node and the estimated cost from that node to the target point. The node with the lowest total cost is then selected as the next node to explore, and the search continues until the target point is reached.

The heuristic function of the A* algorithm is defined as Eq. (1):

(1) f(n)=g(n)+h(n).

In the equation, n represents the currently explored node, f(n) denotes the total cost, g(n) represents the cost incurred from the start node to the current node, and h(n) indicates the estimated cost from the current node to the target point.

Common methods for calculating the estimated cost h(n) include the Manhattan distance, Euclidean distance, and Chebyshev distance, which are expressed mathematically as Eqs. (2)–(4):

(2) h(n)=|x2−x1|+|y2−y1|

(3) h(n)=(x2−x1)2+(y2−y1)2

(4) h(n)=max(|x2−x1|,|y2−y1|).

The choice of heuristic function has a significant impact on the performance of the A* algorithm, making the selection of the estimated cost h(n) critically important. The Euclidean distance effectively approximates the actual distance from the current node to the target node, guiding the algorithm in the direction of the target. Furthermore, utilizing the Euclidean distance allows the paths generated by the algorithm to more closely align with practical application scenarios. Therefore, the improved A* algorithm proposed in this article employs the Euclidean distance function.

Improved A* algorithm

Neighborhood improvement

Neighborhood expansion involves the outward extension of nodes from the current node in various directions. The traditional A* algorithm commonly employs an eight-directional neighborhood expansion method, wherein nodes are expanded in eight different directions from the current node, with each direction separated by an angle of 45°. The schematic representation of this principle is illustrated in Fig. 2A, and the set of coordinates can be expressed as Eq. (5):

Figure 2 Neighborhood extension.

(A) Eight-directional extension, (B) Twenty-four-directional extension.

(5) Tn={(xn±1,yn),(xn,yn±1),(xn±1,yn±1)}.

In the equation, xn represents the x-coordinate of the current search node n, and yn represents the y-coordinate of the current search node n.

Due to the eight-directional search method, the algorithm only expands to eight neighboring nodes at a time. In environments with few obstacles, this can lead to low search efficiency and an excessive number of expanded nodes. To address this issue, we propose a twenty-four directional search method. This method extends the search range from one layer of nodes surrounding the current node to two layers, allowing the algorithm to search towards 24 different nodes simultaneously. The schematic representation of this method is illustrated in Fig. 2B, and the set of coordinates can be expressed as Eq. (6):

(6) Tn={(xn±1,yn),(xn,yn±1),(xn±1,yn±1),(xn±2,yn),(xn,yn±2),(xn±2,yn±2),(xn±2,yn±1),(xn±1,yn±2)}.

While the twenty-four directional search method allows for the exploration of multiple nodes simultaneously (Baoxia, Miaochi & Yong, 2018; Zhimin et al., 2023), its search efficiency can significantly diminish when there are obstacles present within the first layer of neighboring nodes. When the algorithm extends to the second layer of nodes, it is necessary to first assess whether any of the first-layer nodes it traverses contain obstacles. Moreover, the multi-point search approach can result in the algorithm exploring an excessive number of useless nodes, which further reduces its overall efficiency.

In response to the aforementioned issues, this article proposes a novel sixteen-directional extension method. The schematic representation of this method is illustrated in Fig. 3, and the set of coordinates can be expressed as Eq. (7):

Figure 3 New sixteen-directional extension.

(7) Tn={(xn±2,yn),(xn,yn±2),(xn±2,yn±2),(xn±2,yn±1),(xn±1,yn±2)}.

The new sixteen-directional extension method is based on traditional eight-directional and twenty-four-directional extension methods. By removing the eight neighboring nodes surrounding the first layer of nodes in the twenty-four directional framework, the newly formed sixteen-directional neighborhood is established. When there are no obstacles in the eight neighboring nodes of the current node, the algorithm can directly explore the second layer of nodes within the new sixteen-directional neighborhood.

The specific search process of the algorithm is founded on eight-directional exploration, wherein the presence of obstacles in the eight neighboring nodes of the current node is evaluated. If obstacles are found, the algorithm continues to employ the eight-directional extension method. Conversely, if no obstacles exist within the current node’s eight neighboring nodes, the new sixteen-directional extension method is utilized for further exploration.

This hybrid search method determines the use of either the eight-directional extension or the new sixteen-directional extension based on the presence of obstacles in the eight neighboring nodes of the current node. This approach enables effective cross-node exploration, significantly enhancing the search efficiency of the algorithm while reducing the expansion of unnecessary nodes. Figure 4 illustrates a scenario where obstacles are present within the eight-directional neighborhood, with the black grid representing obstacles.

Figure 4 Presence of obstacles in eight-directional neighborhood.

Node optimization

The improved hybrid search method utilizing the eight-directional and new sixteen-directional neighborhoods generates search nodes as illustrated in Fig. 5. During the exploration process from the current node to the target node, both the eight-directional and new sixteen-directional searches may yield invalid nodes. The existence of these invalid nodes results in a waste of resources during the algorithm’s search. To enhance the search efficiency of the algorithm, this article proposes the removal of unnecessary nodes that do not impact the search results.

Figure 5 Expanded nodes.

(A) Eight-directional nodes, (B) Sixteen-directional nodes.

Assuming the angle between the line connecting the current node and the target node and the positive direction of the x-axis is denoted as α, invalid nodes can be eliminated based on the range of angle α. The specific implementation method is as follows:

When employing the eight-directional search method, as shown in Table 1, three unnecessary nodes are removed based on the range of angle α, while five valid nodes are retained.

Table 1 Node selection table for eight-directional search method.

Angle α	Retained direction	Removed direction	
22.5°–67.5°	1,2,3,7,8	4,5,6	
67.5°–112.5°	1,2,3,4,8	5,6,7	
112.5°–157.5°	1,2,3,4,5	6,7,8	
157.5°–202.5°	2,3,4,5,6	1,7,8	
202.5°–247.5°	3,4,5,6,7	1,2,8	
247.5°–292.5°	4,5,6,7,8	1,2,3	
292.5°–337.5°	1,5,6,7,8	2,3,4	
337.5°–360°U0°–22.5°	1,2,6,7,8	3,4,5	

When utilizing the new sixteen-directional search method, as illustrated in Table 2, nine unnecessary nodes are eliminated based on the range of angle α, while seven valid nodes are retained.

Table 2 Node selection table for new sixteen-directional search method.

Angle α	Retained node	Removed node	
11.25°–33.75°	1,2,3,13,14,15,16	4,5,6,7,8,9,10,11,12	
33.75°–56.25°	1,2,3,4,14,15,16	5,6,7,8,9,10,11,12,13	
56.25°–78.75°	1,2,3,4,5,15,16	6,7,8,9,10,11,12,13,14	
78.75°–101.25°	1,2,3,4,5,6,16	7,8,9,10,11,12,13,14,15	
101.25°–123.75°	1,2,3,4,5,6,7	8,9,10,11,12,13,14,15,16	
123.75°–146.25°	2,3,4,5,6,7,8	1,9,10,11,12,13,14,15,16	
146.25°–168.75°	3,4,5,6,7,8,9	1,2,10,11,12,13,14,15,16	
168.75°–191.25°	4,5,6,7,8,9,10	1,2,3,11,12,13,14,15,16	
191.25°–213.75°	5,6,7,8,9,10,11	1,2,3,4,12,13,14,15,16	
213.75°–236.25°	6,7,8,9,10,11,12	1,2,3,4,5,13,14,15,16	
236.75°–258.75°	7,8,9,10,11,12,13	1,2,3,4,5,6,14,15,16	
258.75°–281.25°	8,9,10,11,12,13,14	1,2,3,4,5,6,7,15,16	
281.25°–303.75°	9,10,11,12,13,14,15	1,2,3,4,5,6,7,8,16	
303.75°–326.25°	10,11,12,13,14,15,16	1,9,2,3, 4,5,6,7,8	
326.26°–348.75°	1,11,12,13,14,15	2,3,4,5,6,7,8,9,10	
348.75°–360°U0°–11.25°	1,2,12,13,14,15	3,4,5,6,7,8,9,10,11	

Bidirectional search

Although the traditional A* algorithm has significantly improved its search speed compared to other heuristic algorithms, it still exhibits relatively slow performance, particularly in large-scale map environments, which leads to a noticeable decrease in search efficiency. To address this issue, this article employs a bidirectional search strategy aimed at reducing the number of nodes traversed by the algorithm and enhancing its search speed. The bidirectional search is divided into forward search and backward search. After determining the coordinates of the start and end points, the forward search proceeds from the start point toward the end point, while the backward search goes from the end point back to the start point. The current node of the algorithm is defined as the parent node (Pnode), and the neighboring search nodes are defined as child nodes (Snode). The specific search process of the algorithm is outlined as follows:

Step 1: Initialize the forward search’s Openlist1 and Closelist1, adding the start point to Closelist1 and defining it as Pnode1. Also, initialize the backward search’s Openlist2 and Closelist2, adding the end point to Closelist2 and defining it as Pnode2.

Step 2: Define the neighboring nodes surrounding Pnode1 as SPnode1 and add them to Openlist1. Compute the total cost values of all nodes in Openlist1 using the cost calculation function. Select the node with the minimal cost value and define it as SPnode2, then remove it from Openlist1 and add it to Closelist1.

Step 3: Define the neighboring nodes surrounding Pnode2 as SRnode1 and add them to Openlist2. Similarly, compute the total cost values of all nodes in Openlist2 using the cost calculation function. Select the node with the minimal cost value and define it as SRnode2, then remove it from Openlist2 and add it to Closelist2.

Step 4: Check whether SPnode2 exists in Openlist2 and whether SRnode2 exists in Closelist1. If either exists, it indicates that the forward and backward searches have intersected at a point, and the algorithm concludes. If neither exists, redefine SPnode2 as the new parent node Pnode1 and SRnode2 as the new parent node Pnode2. Steps 2 and 3 are then repeated until an intersection point is found.

Dynamic evaluation function

During bidirectional search, when the start and end points are situated at a considerable distance from one another, there is a possibility that the forward search and backward search may either intersect along a convoluted path or fail to intersect altogether. This situation can lead to a reduction in both the efficiency and accuracy of the algorithm. To address this issue, this article employs a dynamic weight coefficient w(n) to adjust the heuristic function, dynamically balancing the ratio between g(n) and h(n), while controlling the search speed and range of the algorithm.

With the introduction of the dynamic weight coefficient w(n), the total cost calculation for the A* algorithm can be expressed as Eq. (8):

(8) f(n)=g(n)+w(n)⋅h(n).

In the case of forward search, the dynamic weight coefficient is represented as Eq. (9):

(9) w(n)=e|xt−xg|+|yt−yg|⋅(ln|xt−xg|+|yt−yg||xs−xg|+|ys−yg|+1).

In the case of backward search, the dynamic weight coefficient is represented as Eq. (10):

(10) w(n)=e|xu−xg|+|yu−yg|⋅(ln|xu−xg|+|yu−yg||xs−xg|+|ys−yg|+1).

Here, (xs,ys) are the coordinates of the start point, (xg,yg) are the coordinates of the end point, (xt,yt) are the current coordinates in the forward search, and (xu,yu) are the current coordinates in the backward search.

The dynamic weight coefficient combines exponential and logarithmic functions. When the current node is relatively far from the target endpoint, the search speed is accelerated. Conversely, when the current node is closer to the target endpoint, the search speed is reduced to enhance search accuracy and minimize exploration of ineffective areas. This approach improves search efficiency and ensures that the forward and backward searches converge at an intermediate node.

Redundant waypoint deletion strategy

Even after the A* algorithm has planned the optimal path, a number of redundant waypoints may still exist. To address this issue, this article proposes a waypoint deletion strategy that effectively removes unnecessary waypoints by assessing whether obstacles exist between nodes. As illustrated in Fig. 6, the specific implementation method is as follows:

Figure 6 Deletion of redundant waypoints.

Step 1: Starting from the endpoint P10, connect it to the adjacent waypoint P8 and check for the presence of obstacles along the connecting line. If no obstacles are present, the waypoint P9 can be deleted.

Step 2: Connect P10 to P7 and examine whether there are obstacles along this line. If obstacles are detected, waypoint P8 is designated as a non-deletable waypoint. If no obstacles are found, waypoint P8 can be removed.

Step 3: Use P8 as the new starting waypoint and repeat Steps 1 and 2 until reaching the starting point P1. All non-deletable waypoints are then reconnected to form the new optimal path.

Overall process of the improved algorithm

In summary, the overall flowchart of the improved A* algorithm is illustrated in Fig. 7.

Figure 7 Overall flowchart of the algorithm.

In order to help readers understand the process of the algorithm more intuitively, we use pseudocode to describe the process of the entire algorithm, and the specific algorithm pseudocode is as follows:

 # 1. define node class	
 class Node:	
  def __init__(self, position, parent=None):	
 # 2. Define heuristic functions	
 def heuristic(node, goal):	
 # 3. Get a neighbor node	
 def get_neighbors(node, grid, goal, mode):	
 # 4. Define a two-way A* search	
 def a_star_bidirectional(start, goal, grid, connectivity = 8):	
 # 5. Check if the paths meet and merge the paths	
 def meet_in_the_middle():	
 # 6. Read map data from an Excel file	
 def read_data_from_excel(file_path):	
 # 7. Optimize the path	
 def optimize_path(path, grid, goal):	
 # 8. Draw maps and routes	
 def plot_grid(grid, search_start, search_goal, path1, path2, start, goal):	
 # Main process	
 file_path = ‘map_model.xlsx’	
 grid, start, goal = read_data_from_excel(file_path)	
 path1, path2, search_path_start, search_path_goal = a_star_bidirectional(start, goal, grid)	
 optimized_path = optimize_path(path1 + path2, grid, goal)	
 plot_grid(grid, search_path_start, search_path_goal, optimized_path, [], start, goal)	

Simulation experiment analysis

Experimental environment

To evaluate the performance of the proposed algorithm, we constructed a simulation environment and conducted simulation experiments. All experiments were executed on a computer equipped with an Intel Core i9 processor, 128 GB of RAM, and two NVIDIA RTX 4090 GPUs with 24 GB of video memory each, utilizing the Python programming language for implementation.

Comparison experiment of improved A* algorithm 1

First, to demonstrate the advantages of the improved neighborhood search mechanism proposed in this article, comparative experiments were conducted against the four-neighborhood and eight-neighborhood A* algorithms. Two different environmental maps were created randomly: a 30 × 30 grid and a 50 × 50 grid. In the 30 × 30 map, the starting point and endpoint were set at (1, 1) and (30, 30), respectively, while in the 50 × 50 map, the starting point and endpoint were set at (1, 1) and (50, 50), respectively. In this experiment, only the improved algorithm in “Neighborhood Improvement” and “ Node Optimization” is verified, which is called “Improved A* Algorithm 1”.

The simulation results are shown in Figs. 8 and 9, where the black grids represent obstacles, the gray grids represent expanded nodes, the green grids represent path nodes, and the yellow lines indicate the planned path. Tables 3 and 4 present the algorithm’s performance parameters under different map environments.

Figure 8 Comparison of algorithm effectiveness on the 30 × 30 grid map.

(A) Four-neighborhood A* Algorithm, (B) Eight-neighborhood A* Algorithm, (C) Improved A* Algorithm 1.

Figure 9 Comparison of algorithm effectiveness on the 50 × 50 grid map.

(A) Four-neighborhood A* Algorithm, (B) Eight-neighborhood A* Algorithm, (C) Improved A* Algorithm 1.

Table 3 Experimental data in the 30 × 30 grid map environment.

Algorithm	Number of expanded nodes	Path length	Number of path points	Time (s)	
Four-neighborhood A* algorithm	317	58	59	0.306	
Eight-neighborhood A* algorithm	114	42.18	32	0.327	
Improved A* algorithm 1	98	42.18	28	0.001	

Table 4 Experimental data in the 50 × 50 grid map environment.

Algorithm	Number of expanded nodes	Path length	Number of path points	Time (s)	
Four-neighborhood A* algorithm	861	98	99	2.21	
Eight-neighborhood A* algorithm	206	75.74	61	2.075	
Improved A* algorithm 1	182	71.10	38	0.0029	

By comparing Figs. 8A and 8C, it can be observed that the Improved A* Algorithm 1 demonstrates significant enhancements in the neighborhood expansion of nodes compared to the four-neighborhood A* algorithm in different map environments. Concurrently, the algorithm’s search efficiency has markedly increased. Furthermore, by comparing Figs. 9B and 8C, it is evident that the Improved A* Algorithm 1 optimizes neighborhood expansion nodes in relation to the eight-neighborhood A* algorithm across different map environments, along with an improvement in search efficiency.

Additionally, as shown in Tables 3 and 4, the performance indicators of the proposed Improved A* Algorithm 1 exhibit enhancements relative to the four-neighborhood A* algorithm, with the number of expanded nodes reduced by 69.08% and 78.86%, respectively. In comparison with the eight-neighborhood A* algorithm, while the path lengths remain relatively unchanged, the number of expanded nodes was decreased by 14.03% and 11.65%, respectively. The significant reduction in the number of expanded nodes effectively validates the efficacy of the neighborhood improvement strategy proposed in this article.

Comparative experiment of the improved A* Algorithm 2

To validate the effectiveness of the proposed improved A* algorithm (All improvement methods in “Problem Description”, referred to as “Improved A* Algorithm 2”), comparative experiments were conducted between the Improved A* Algorithm 2 and both the four-neighborhood A* algorithm and the eight-neighborhood A* algorithm in 64 × 64 and 128 × 128 grid map environments. The simulation environment was constructed as a two-dimensional grid map, where black grids represent obstacles, gray grids indicate forward expansion nodes, orange grids signify backward expansion nodes, green grids denote path nodes, and yellow lines illustrate the planned path. In the 64 × 64 map, the starting point and the endpoint are set at coordinates (1, 1) and (64, 64) respectively; whereas, in the 128 × 128 map environment, the starting point and endpoint are set at (1, 1) and (128, 128).

The simulation results are presented in Figs. 10 and 11, with the data comparisons delineated in Tables 5 and 6 below.

Figure 10 Algorithm performance comparison in the 64 × 64 grid map.

(A) Four-neighborhood A* Algorithm, (B) Eight-neighborhood A* Algorithm, (C) Improved A* Algorithm 2.

Figure 11 Algorithm performance comparison in the 128 × 128 grid map.

(A) Four-neighborhood A* Algorithm, (B) Eight-neighborhood A* Algorithm, (C) Improved A* Algorithm 2.

Table 5 Experimental data in the 64 × 64 grid map environment.

Algorithm	Number of expanded nodes	Path length	Number of path points	Time (s)	
Four-neighborhood A* algorithm	2,506	126	127	0.505	
Eight-neighborhood A* algorithm	308	97.29	78	0.325	
Improved A* algorithm 2	251	92.18	29	0.003	

Table 6 Experimental data in the 128 × 128 grid map environment.

Algorithm	Number of expanded nodes	Path length	Number of path points	Time (s)	
Four-neighborhood A* algorithm	5,125	254	255	1.014	
Eight-neighborhood A* algorithm	602	188.39	143	0.40	
Improved A* algorithm 2	525	182.54	53	0.0079	

By comparing the experimental results presented in Figs. 10 and 11, it can be observed that the proposed Improved A* Algorithm 2 significantly reduces the number of expanded nodes and path points, shortens the path length, and enhances efficiency in various map environments compared to the four-neighborhood A* algorithm and the eight-neighborhood A* algorithm. As shown in Table 5, in the 64 × 64 grid map environment, the Improved A* Algorithm 2 reduces the number of expanded nodes by 89.98% and 18.5% compared to the four-neighborhood and eight-neighborhood A* algorithms, respectively. The path length is shortened by 26.84% and 5.25%, respectively, while the number of path points decreases by 77.16% and 62.82%, and the search time is reduced by 99.4% and 99.07%. Table 6 indicates that in the 128 × 128 grid map environment, the Improved A* Algorithm 2 reduces the number of expanded nodes by 89.75% and 12.79% compared to the four-neighborhood and eight-neighborhood A* algorithms, respectively. The path length is shortened by 28.13% and 3.1%, while the number of path points decreases by 79.21% and 62.93%, and the search time is reduced by 99.22% and 98.02%. These results demonstrate that the performance indicators of the Improved A* Algorithm 2 surpass those of both the four-neighborhood and eight-neighborhood A* algorithms in all aspects, significantly decreasing the number of expanded nodes and path points, thereby greatly enhancing the search efficiency of the algorithm while also shortening the path length. This further validates the effectiveness and superiority of the Improved A* Algorithm 2.

Conclusion

This article presents an improved A* algorithm targeting the path planning problem of mobile robots. Initially, the traditional eight-neighborhood and twenty-four-neighborhood expansions are optimized into a new sixteen-neighborhood model. A hybrid neighborhood expansion method combining eight-neighborhood and sixteen-neighborhood approaches is proposed. The decision to utilize eight-neighborhood search or the new sixteen-neighborhood search is determined by checking for obstacles within the current node’s eight-neighborhood. Subsequently, unnecessary search nodes are eliminated based on neighborhood direction search rules, effectively reducing the number of expanded nodes and enhancing the efficiency of the algorithm.

Furthermore, an improved bidirectional search mechanism is introduced, employing dynamic weight coefficients to further increase the search efficiency of the algorithm. Additionally, a strategy for extracting key nodes is implemented to effectively eliminate redundant waypoints and significantly address the issue of superfluous nodes.

Finally, comprehensive simulations are conducted under various map environments, and the experimental data demonstrate that the proposed improved A* algorithm excels in terms of the number of expanded nodes, the count of path points, path length, and search time. These results validate the effectiveness of the improved A* algorithm in addressing the path planning problem for mobile robots.

Supplemental Information

Supplemental Information 1 the code files of the paper.

Additional Information and Declarations

Competing Interests

The authors declare that they have no competing interests.

Author Contributions

Xing Fu conceived and designed the experiments, performed the experiments, performed the computation work, authored or reviewed drafts of the article, and approved the final draft.

Zucheng Huang conceived and designed the experiments, performed the experiments, analyzed the data, prepared figures and/or tables, and approved the final draft.

Gongxue Zhang performed the computation work, authored or reviewed drafts of the article, and approved the final draft.

Weijun Wang analyzed the data, authored or reviewed drafts of the article, and approved the final draft.

Jian Wang performed the experiments, prepared figures and/or tables, and approved the final draft.

Data Availability

The following information was supplied regarding data availability:

The code are available in the Supplemental Files.

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
