# Peer review of "Research on path planning of mobile robots based on improved A* algorithm"

_PeerJ Computer Science, doi:10.7717/peerj-cs.2691_

## Round 0.1 · original submission · Major Revisions

· Academic Editor

Major Revisions

The reviews are generally positive, but have revealed some problems. Please address all the issues raised by the reviewers.

Reviewer 1 ·

Basic reporting

-English language usage: Although mostly clear, there are some grammatical errors and awkward phrasings throughout the paper. A thorough proofreading by a native English speaker or professional editing service would enhance readability and professionalism.
-Literature review: The background provided is adequate, but could be more comprehensive. Including more recent advancements in A* algorithm improvements for robot path planning would strengthen the context and demonstrate the novelty of the proposed approach.
-Figures and tables: While relevant, some figures (particularly Figures 2-5) would benefit from more detailed captions and explanations to help readers better understand the proposed improvements.
-Raw data: It's not clear if the raw data from the experiments has been shared. Including this data or providing a link to a repository would enhance reproducibility.
-Definitions and proofs: While the article includes formal results, some of the design choices (e.g., specific angles used in node selection tables) could use more thorough explanations or mathematical justifications.

Suggested improvements:

1.Conduct a thorough language review to improve grammar and clarity.
2.Expand the literature review to include more recent related works.
3.Enhance figure captions and explanations, particularly for Figures 2-5.
4.Ensure raw experimental data is shared or easily accessible.
5.Provide more detailed explanations or proofs for key algorithm design choices.

Regarding the literature review, I strongly recommend including references to the following recent and relevant research:
-Rybczak, M., Popowniak, N., & Lazarowska, A. (2024). A survey of machine learning approaches for mobile robot control. Robotics, 13(1), 12.

While your work doesn't focus on machine learning directly, referencing this survey paper would provide a broader context of mobile robot control and could offer directions for future research developments.
-Promkaew, N., Thammawiset, S., Srisan, P., Sanitchon, P., Tummawai, T., & Sukpancharoen, S. (2024).

Development of metaheuristic algorithms for efficient path planning of autonomous mobile robots in indoor environments. Results in Engineering, 22, 102280.

This research presents the development of metaheuristic algorithms for path planning of autonomous mobile robots in indoor environments, which is directly relevant to your work. Referencing this paper would help show the latest advancements in path planning and allow for a comparison of your proposed method with metaheuristic approaches.

Including these references would strengthen your literature review, demonstrate awareness of the latest research in the field, and help readers understand how your work relates to and advances upon current state-of-the-art methods. It also provides an opportunity to compare your approach with other existing methods, highlighting the importance and innovation of your research.

Experimental design

While the experimental design of the article generally meets the required standards, there are a few areas that could be improved:

1. Knowledge gap clarification: Although the research question is well-defined, the authors could more explicitly state how their work fills a specific knowledge gap in the field of mobile robot path planning. A clearer statement of the novelty and significance of their approach in relation to existing literature would strengthen the paper.

2. Experimental setup details: While the methods are described with sufficient detail for replication, more information on the specific hardware and software configurations used for the simulations would be beneficial. This would enhance reproducibility and allow other researchers to more accurately compare their results.

3. Parameter justification: The authors should provide more thorough justifications for the chosen parameters in their improved A* algorithm, particularly for the angles used in the node selection tables. A more detailed explanation of how these choices were made would add rigor to the experimental design.

4. Comparative baseline: While the authors compare their improved algorithm to traditional A* variants, it would be valuable to include comparisons with other state-of-the-art path planning algorithms. This would provide a more comprehensive evaluation of the proposed method's performance.

5. Robustness testing: The experimental design could be strengthened by including tests on a wider variety of environments, including more complex and dynamic scenarios. This would demonstrate the robustness and generalizability of the proposed algorithm.

6. Statistical analysis: The addition of statistical tests to validate the significance of the performance improvements would enhance the rigor of the investigation.

Suggested improvements:
1. Clearly articulate the specific knowledge gap being addressed and how this research fills it.
2. Provide more detailed information on the simulation environment and hardware specifications.
3. Offer more thorough justifications for algorithm parameter choices.
4. Include comparisons with additional state-of-the-art path planning algorithms.
5. Expand the range of testing environments to demonstrate robustness.
6. Incorporate statistical analysis to validate the significance of the results.

Validity of the findings

While the article generally meets the standards for validity of findings, there are several areas that could be improved:

1. Statistical analysis: Although the underlying data appears to be provided, the statistical soundness of the results could be strengthened. The authors should consider including measures of variability (e.g., standard deviations) and conducting statistical tests to demonstrate the significance of the performance improvements.

2. Replication details: While the authors encourage replication, they could provide more explicit instructions or guidelines for other researchers wishing to replicate or build upon their work. This could include sharing code repositories or providing more detailed experimental protocols.

3. Robustness of findings: The authors should consider testing their algorithm on a wider range of scenarios or environments to demonstrate the robustness and generalizability of their findings. This would strengthen the validity of their conclusions.

4. Limitations discussion: The paper would benefit from a more thorough discussion of the limitations of the proposed approach. Identifying potential edge cases or scenarios where the algorithm might not perform as well would provide a more balanced view of the findings.

5. Comparative analysis: While the conclusions are well-stated and linked to the original research question, the paper could benefit from a more comprehensive comparison with other state-of-the-art algorithms beyond just the traditional A* variants.

6. Long-term implications: The authors could expand on the potential long-term implications of their findings for the field of mobile robot path planning. This would help readers better understand the broader impact of the research.

Suggested improvements:
1. Include more rigorous statistical analysis of the results.
2. Provide more detailed guidelines for replication of the study.
3. Expand testing to a wider range of scenarios to demonstrate robustness.
4. Add a comprehensive discussion of the limitations of the proposed approach.
5. Include comparisons with additional state-of-the-art algorithms.
6. Discuss the broader implications and potential future directions of the research.

These enhancements would further strengthen the validity of the findings and increase the overall impact and contribution of the research to the field.

Cite this review as

·

Basic reporting

• To make the manuscript title more suitable, I suggest changing it to "Research on path planning of mobile robots based on improved A* algorithms".
• The most significant results should be included in the abstract.
• An appropriate reference is necessary for all equations. Please mention the reference number for each equation.
• In lines 42-44, the authors define the primary goal of path planning as "the primary goal of path planning is to swiftly and accurately find an optimal collision-free path from a starting position to a target position in an unknown environment," where the path planning is used in both an unknown environment and a known environment (as the used environment in this manuscript).
• In lines 45-51, the authors classify the path planning into traditional and intelligent algorithms, but the distinction between global, local, and intelligent algorithms could need further clarification. For example, while intelligent algorithms can be used for global path planning, they are often not as commonly implemented in real-time scenarios compared to algorithms like A* or DWA. Also, RRT is primarily a global planner but may require fine tuning to be optimal, especially in dynamic environments. Please use Reference [1] for more information. Also, the reference 14 is not focused on the paper topics; please check that.

Experimental design

• In Section 3.2, Node Optimization, invalid nodes are eliminated based on alpha angle constraints as outlined in Tables 1 and 2. Should the remaining nodes after this filtering process all represent obstacles, the mobile robot must employ alternative pathfinding strategies. Also, to clarify the node elimination process, a visual representation of Table 1 is provided in a figure. Nodes are numbered as indicated in the table, allowing for a more intuitive understanding of the alpha angle-based filtering criteria.

Validity of the findings

• The sixteen-neighborhood search, introduced as a compromise between the eight-neighborhood and twenty-four-neighborhood searches, represents a novel concept. However, the paper does not address the computational complexity introduced by the additional mechanisms. A discussion on whether the computational cost is negligible in comparison to the benefits would enhance the credibility of the claims.

Reviewer 3 ·

Basic reporting

The article uses appropriate grammar to describe the results, but the logic between Part 2 and Part 3 needs to be more clear. For example, line 146 mentions a heuristic function, which is not mentioned in any previous equation or paragraph. Furthermore, the formula editor should be used to edit all formulas and letters in the article, not just when editing formulas. This kind of problem appears rapidly in the article.

Overall, the literature references provides a sufficient background, however, aside from the background part, some theories in section 3 still need some references to prove, for example, in line146-152.

The figures and tables shared the raw data properly, and authors analysed the results.

Experimental design

The design of the experiment is relatively clear and the levels are clear, highlighting the results that the author wants to show.
We can find out some of their research question through the article, but I think they need to give one section to the research questions to make their directions more clear.
They provide some details to replicate their methods, but I think they could provide readers a pseudocode to make their methods more clear.

Validity of the findings

The results is one of the big problem of this article.
First, in their experimental results, they only provide readers with a single set of data based on their research steps. It seems that the authors did not conduct multiple sets of controlled repeated experiments to verify their results. In the analysis, they easily made conclusions based only on a set of data and images provided, which I think is undesirable. Especially, compared with the traditional A* algorithms, they intentionally reduced the search area and skipped some steps to improve search efficiency. At the same time, they also proposed that in their two-way search method, there may be a problem that the forward and reverse searches cannot merge. In order to verify the effectiveness of their method, they need to provide a variety of control data to prove the effectiveness of their method.

Secondly, in the search part, they did not compare with the 24-way search method. Since the authors claim that they have made some improvements based on the 8 and 24-direction search methods, they must compare the improved method with the 24-direction search method.

In addition, they need more data to demonstrate that their data has some improvement over traditional A* search methods, especially their search efficiency mentioned in the previous section.

All in all, I think the authors need to conduct more experiments on the finding part to prove the effectiveness of their method.

Cite this review as

---

## Round 0.2 · Major Revisions

· Academic Editor

Major Revisions

As you can see there are still some issues raised by the reviewers. Please address all of these before publication.

·

Basic reporting

no comment

Experimental design

no comment

Validity of the findings

no comment

Reviewer 3 ·

Basic reporting

1. The language usage is mostly clear, however, there are still many problems with the format of the article. For example, symbols such as h(n) in the paragraph need to be edited using the formula editor instead of simply typing. I have pointed this out before but you still haven't made any improvements.

2. I still recommend that you provide some pseudocode in the article. Although I know you have provided the original code, not everyone will spend a lot of time verifying the feasibility of your code. Pseudocode can make your ideas more clear in the article and provide convenience to readers.

3. I have read your response, however, we still need to know whether your algorithm have any limitations. For example whether the algorithm is suitable for all environments, if it surpasses other algorithms in all aspects, or are there any exceptions, etc.

Experimental design

Authors should state in their articles that their maps are randomly generate.

Validity of the findings

1. In your response, comparing with the 24-neighborhood algorithm is one of your future research goals. However, from lines 177-182, you mentioned the shortcomings of the 24-neighbor algorithm, but did not give any references or data to support it. Meanwhile, from my point of view, your article is based on 8-neighborhood and 24-neighborhood algorithms. The comparison with the 24-neighborhood algorithm should be part of your article to provide evidence for the superiority of your method rather than a future research direction.

2. In the figure results part, you can use single experimental result as an example to show the advantages and I know you have shared your raw data. However, in the data results part, what you need to provide is the average result after multiple random experiments, such as the average number of nodes explored, average path length, etc. after 50-100 repeated experiments on a random map, rather than the conclusion of a single experiment.

Cite this review as

---

## Round 0.3 · accepted · Accept

· Academic Editor

Accept

You have have addressed all the issues raised by the reviewers. You paper is ready for publication.

Reviewer 3 ·

Basic reporting

The revised article has shown significant improvements in both content and format, given the improvements they made, I recommend accepting the manuscript for publication.

Experimental design

N/A

Validity of the findings

N/A

Cite this review as